# Interfering with the Ubiquitin-Mediated Regulation of Akt as a Strategy for Cancer Treatment

**DOI:** 10.3390/ijms24032809

**Published:** 2023-02-01

**Authors:** Elena Paccosi, Alessio Balzerano, Luca Proietti-De-Santis

**Affiliations:** Unit of Molecular Genetics of Aging, Department of Ecology and Biology, University of Tuscia, 01100 Viterbo, Italy

**Keywords:** Akt, ubiquitination, cancer

## Abstract

The serine/threonine kinase Akt modulates the functions of numerous substrates, many of them being involved in cell proliferation and growth, metabolism, angiogenesis, resistance to hypoxia and migration. Akt is frequently deregulated in many types of human cancers, its overexpression or abnormal activation being associated with the increased proliferation and survival of cancer cells. A promising avenue for turning off the functionality of Akt is to either interfere with the K63-linked ubiquitination that is necessary for Akt membrane recruitment and activation or increase the K48-linked polyubiquitination that aims to target Akt to the proteasome for its degradation. Recent evidence indicates that targeting the ubiquitin proteasome system is effective for certain cancer treatments. In this review, the functions and roles of Akt in human cancer will be discussed, with a main focus on molecules and compounds that target various elements of the ubiquitination processes that regulate the activation and inactivation of Akt. Moreover, their possible and attractive implications for cancer therapy will be discussed.

## 1. The Akt Signaling Pathway in Cancer

The PI3K/Akt/mTOR pathway has been shown to be aberrantly activated in many human cancers, including breast, lung, ovarian, pancreatic and gastric carcinomas [1]. In all these cases, Akt is responsible for driving cell growth and tumor progression [2,3,4,5,6]. Indeed, phosphorylated Akt (pAkt) translocates to a plethora of intracellular locations, where it modulates the function of numerous substrates, many of them being involved in cell proliferation and growth, metabolism, angiogenesis, resistance to hypoxia and migration [7,8] (Figure 1). All three Akt isoforms (Akt1, Akt2 and Akt3) were found to be overexpressed in human cancers [9,10,11,12,13,14,15,16,17,18,19], where, besides promoting tumor progression, they are also involved in conferring acquired resistance to many conventional chemotherapeutic agents [20,21,22], mainly for their contribution to the Epithelial-Mesenchymal Transition (EMT) occurring in drug-resistant and metastatic cancer cells [23,24,25]. In this regard, it is worth noting that either the overexpression or amplification of Akt1 and Akt2 is associated with ovarian cancer cells’ acquired resistance to paclitaxel [26]. The Cancer Genome Atlas (TCGA) pan-cancer datasets have profiled a comprehensive catalog of PI3K/Akt/mTOR-associated variants across 10,000 tumors and represent a valuable resource for both understanding the PI3K/Akt/mTOR pathway deregulation in cancers and translating this information in clinical utility for personalized treatments [27,28,29]. Although the Akt pathway is unquestionably believed to be a gold target for cancer therapy, to date, no drugs targeting Akt have been approved for any cancer treatments. For this reason, the discovery of a K63-linked ubiquitination of Akt, specifically involved in its activation, has focused the attention on the appealing opportunity of studying this ubiquitin-mediated regulation as a druggable target in future molecular investigations.

### 1.1. Akt: An Overview

The serine/threonine kinase Akt, alias Protein Kinase B (PKB), is activated by differ-ent kinds of stimuli, including cytokines, hormones, stresses and growth factors like Vascular Endothelial Growth Factor (VEGF), Fibroblast Growth Factor (FGF), Nerve Growth Factor (NGF), Epidermal Growth Factor (EGF), Platelet-Derived Growth Factor (PDGF) and Insulin-like Growth Factor (IGF) [30]. All the Akt isoforms are composed of three domains: an N-terminal Pleckstrin Homology (PH) domain, a central kinase domain separated by a hinge region and a C-terminal regulatory domain. All three Akt kinases show a highly conserved primary structure, sharing 90–95% of the homology in the kinase region and 60% of the homology in their PH domain [31,32]. The main Akt isoform is encoded by the *Akt1* gene; it was shown that each Akt isoform is differentially expressed at both the mRNA and protein levels, and each one exerts different roles not only in a physiological context but also in cancer pathogenesis [33]. Studies of Akt isoforms performed in knockout mice have documented that the different Akt isoforms may exert tissue-specific and not completely overlapping functions; indeed, *Akt1* knockout mice turned out to be smaller than their wild-type counterparts, indicating that Akt1 has a critical role in the tight regulation of cell survival and cell growth [34]; *Akt2* knockout mice turned out to be prone to developing a type 2 diabetes-like phenotype, suggesting a main role for Akt2 in the maintenance of glucose homeostasis [35]; *Akt3* knockout mice, instead, displayed an impaired brain development, thus demonstrating a major role for Akt3 in neurogenesis [36]. Furthermore, several studies addressed differential roles for Akt1 and Akt2 in the regulation of cell cycle progression and cell migration [37,38].

Akt kinase activity is finely tuned by post-translational modifications such as phosphorylation, ubiquitination, acetylation, palmitoylation, hydroxylation and methylation [39,40,41], which may also be relevant for the Akt hyperactivation observed in cancers [42]. Akt activation requires its translocation to the cell membrane, which is mediated by the binding of the PH domain to the phosphatidylinositol 3,4,5-trisphosphate (PIP3) [43]. PIP3 is a phospholipid generated by Phosphatidylinositol 3 Kinase (PI3K), which is able to act as a second messenger for the recruitment to the membrane of a subset of signaling proteins with PH domains, including not only Akt but also the 3-Phosphoinositide-Dependent Protein Kinase 1 (PDK1) [44]. In the OFF state, the PH induces the inhibition of Akt activation, by interacting with the kinase domain [45]. Instead, when the PH domain is engaged by PIP3, it triggers the membrane translocation of Akt and its subsequent phosphorylation by PDK1 and mTOR complex 2 (mTORC2) [46]. Indeed, the binding of PIP3 to the PH domain of Akt causes a conformational change which makes Akt prone for the PDK1-mediated phosphorylation of T308. Even if the phosphorylation mediated by PDK1 is pivotal, it is still not sufficient for Akt activation; indeed, a second phosphorylation at S473 is required, mediated by the kinase mammalian Target Of Rapamycin (mTOR), which acts as a part of two complexes called mTORC1 and mTORC2, the latter one being the responsible kinase for S473 phosphorylation [47]. Besides directly phosphorylating Akt, mTORC2 phosphorylates and activates the IGF receptor, thus indirectly promoting Akt activation [48,49]. mTORC1, instead, up-regulates protein synthesis by phosphorylating the Eukaryotic translation Initiation Factor 4E Binding Protein 1 (EIF4EBP1) and the Ribosomal S6 Kinase family members RPS6KB1 and RPS6KB2, thereby leading to an increased protein synthesis [50]. The above-described PI3K/Akt/mTOR signaling pathway is regulated by the Phosphatase and Tensin homolog (PTEN) protein, a phosphatase that is able to suppress the PI3K/Akt signaling pathway by de-phosphorylating PIP3 (Figure 1) [51,52,53]. Last but not least, Akt activation may also be promoted by a K63-linked ubiquitination. As will be deeply explained in the next chapter, this specific non-degrading ubiquitination acts as a critical step for Akt membrane recruitment and phosphorylation [54,55,56,57].

The Akt kinase can directly phosphorylate a plethora of proteins (to date, more than one hundred Akt substrates have been identified [58], most of them being involved in cell survival, proliferation, migration and metabolism [7,8,9]). The Akt pathway transmits signals from upstream regulatory proteins, such as PTEN and PI3K, to many downstream effectors, such as Glycogen Synthase Kinase 3 (GSK3), Forkhead transcription factor (FOXO) and Murine Double Minute 2 (MDM2) [59,60,61]. Each Akt isoform has been demonstrated to play different and specific roles in cancer cells signaling; indeed, Akt1 and Akt2 displayed opposite roles in cell cycle progression, migration and invasion among different types of human cancers [62,63].

### 1.2. Akt in Proliferation and Cell Survival

Akt may promote cell proliferation and survival by regulating many different targets. First of all, Akt exerts its major oncogenic role by modulating the PI3K/Akt signaling pathway, which is responsible for promoting cell proliferation and survival and preventing apoptosis [64].

When activated by growth factors, IGF and EGF receptors dimerize and undergo autophosphorylation, thus activating a cascade of phosphorylations that leads to PI3K activation and PIP2 conversion to PIP3. PIP3 recruits to the plasma membrane both Akt and PDK1, which phosphorylate Akt. The activated Akt, in turn, induces the release of the inhibitory Tuberous Sclerosis Complex (TSC, composed of TSC1 and 2) from the Ras-related small G protein Rheb-GTPase (Rheb), a step required for the activation of both Rheb and mTORC1 [65]. mTORC1 acts as a regulator of cell survival and growth by stimulating translation initiation and ribosome biogenesis, by activating the transcription of genes involved in cell survival [66] and by catalyzing the phosphorylation of multiple targets, such as ribosomal protein S6 Kinase (p70S6K) and eukaryotic translation initiation factor 4E binding Protein 1 (4E-BP1) (Figure 1) [49].

Akt oncogenic activation confers resistance to apoptosis [67] due to its crosstalk with the tumor suppressor p53. Indeed, the Akt-mediated phosphorylation of MDM2 results in MDM2 import in the nucleus, where it promotes the ubiquitination and consequent proteasomal degradation of p53 [68,69]. Furthermore, the PI3K/Akt pathway activation in tumors may be accompanied by the tumor suppressor PTEN inactivating mutations, thus boosting survival advantages and the uncontrolled proliferation of cancer cells, lacking the PTEN-mediated cell cycle arrest in the G1 phase [70].

Akt may also act as an inhibitor of cell cycle arrest by phosphorylating the tumor suppressor p21; briefly, Akt-mediated phosphorylation induces the cytoplasmic localization of p21 and its sequestration mediated by the bind with 14-3-3 proteins [71,72]. The activation of the Akt pathway may promote the expression of anti-apoptotic proteins such as B-cell Lymphoma 2 (BCL2) [73]. In addiction, Akt has also been found to phosphorylate and inhibit the BCL2-Associated agonist of cell Death protein (BAD) [74], p27 and Bim, the latter two through the phosphorylation and nuclear export of FOXO, which is essential for their transcription, promoting cell proliferation and survival overall [75,76]. Furthermore, Akt promotes cell survival and suppresses apoptotic death by inducing the X-linked inhibitor of apoptosis protein (XIAP) [77]. In addition, Akt directly phosphorylates human procaspase-9, thus leading to a decrease in its protease activity (Figure 1) [78,79].

### 1.3. Akt in Cell Migration and Metastasis

Akt, despite promoting tumor growth, also promotes cell migration and metastasis. Metastasis is a multistep process starting from the loss of cell adhesion, the invasion into local vessels and tissues and, lastly, the colonization of distant sites. Akt plays a key role in both metastasis and invasion, its expression being higher in distant metastasis than in the primary tumor [80].

When activated by Vascular Endothelial Growth Factor (VEGF), Akt plays a key role in the angiogenesis required for tumor growth, promoting endothelial cell survival, growth and proliferation. VEGF may enhance the Akt signaling pathway to regulate the expression of not only VEGF itself and its receptor [81] but also Hypoxia Inducible Factor 1α (HIF-1α), heme oxygenase 1, inducible Nitric Oxide Synthase (iNOS) and several glycolytic enzymes required for angiogenesis induction [82]. In tumors, the sustained activity of Akt stimulates both the endothelial cells migration and the formation of structurally abnormal blood vessels; moreover, it promotes tumor angiogenesis by activating angiopoietins and endothelial nitric oxide [83,84]. Furthermore, Akt up-regulates Transforming Growth Factor-β2 (TGFβ2) expression, thus promoting cancer metastasis [85].

The PI3K/Akt/mTOR signaling pathway plays a critical role in EMT, a hallmark of which is the downregulation of E-cadherin [86]. EMT is a biological process that plays a key role in tumor cell invasion, metastasis and chemoresistance. It can be induced either directly or indirectly by PI3K/Akt/mTOR proteins in cooperation with other signaling pathways, such as TGFβ, NF-kB, Ras and Wnt/β-catenin [23].

The remodeling of the extracellular matrix (ECM) is required for cell migration and invasion. In this process, invadopodia, which are actin-rich structures associated with the plasma membrane, exert a relevant role [87]. Both invadopodia formation and activity are triggered by the PI3K/Akt signaling. In this context, the balance between the formation of PIP3 and PI34P2 PI3Ks has been shown to modulate the metastatic potential of many cancer cell lines [88]. Indeed, in MDA-MB-231 breast cancer cells, the knockdown of the catalytic subunit of PI3K p110α and the knockdown of Akt are able to attenuate invadopodia formation [89].

### 1.4. Akt in Cancer Metabolism Regulation

The PI3K/Akt signaling network has a complex and critical role in cancer metabolism [90,91]. While the metabolic functions of Akt support its physiological functions in cell survival, growth and proliferation, de-regulated Akt function instead supports the abnormal proliferation and survival of cancer cells by deregulating the control of the metabolism.

PI3K/Akt signaling may affect cell metabolism both directly and indirectly through the phosphorylation-mediated regulation of metabolic enzymes or regulators of nutrient transport or through the activation of key downstream effectors involved in cellular metabolic reprogramming, such as the mTORC1, GSK3 and FOXO family members [91,92].

An altered glucose metabolism is a common metabolic hallmark distinguishing cancer cells. In response to growth factors, PI3K/Akt signaling controls nutrient uptake and metabolism, exerting its most relevant role in regulating glucose metabolism by trafficking the cellular uptake of glucose and by inducing gene expression. Akt may alter the expression of Glucose Transporter 1 (GLUT1) by increasing the translation of its mRNA through mTORC1 and 4E-BP1 [93].

GSK3 acts as a negative regulator of glucose metabolism since it is the kinase responsible for inhibiting glycogen synthase by phosphorylating this enzyme at Ser652 [94]. Insulin stimulation reverses the GSK3-mediated phosphorylation, thus promoting glycogen synthesis [95]. Indeed, in response to insulin, Akt, which inhibits both GSK3α and GSK3β [96], may stimulate glycogen synthesis [97].

## 2. Two Distinct Ubiquitination Processes Regulate the Activation and Inactivation of Akt Kinase

Ubiquitin is an 8.6 kDa regulatory protein originally suggested to mark tagged proteins to the degradation via 26S proteasome in a process defined as ubiquitination. Over the years, the role of the ubiquitination has been enlarged to multiple functions, including the trafficking, stabilization and activation of the target protein.

The covalent addition of ubiquitin to the substrate protein is mediated by the concerted action of three enzyme families, called E1, E2 and E3. First, a ubiquitin-activating enzyme (E1) activates the ubiquitin through an ATP-dependent reaction. The activated ubiquitin is hence transferred to a ubiquitin-conjugating enzyme (E2). Finally, a ubiquitin-protein ligase E3 mediates the transfer of the ubiquitin from E2 to a specific lysine side chain of a target protein [98,99]. In the human genome, there are over 500 genes codifying for a specific E3 ligase [100]. The E3 ubiquitin ligase involved in the ubiquitination cascade is crucial to defining the substrate specificity and the spatiotemporal nature of the pathway.

Since the amino acid chain of ubiquitin contains seven lysine residues (K6, K11, K27, K29, K33, K48 and K63), in the poly-ubiquitination process, the ubiquitin molecules can be linked to each other through each of these seven different sites. The diverse ways to assemble ubiquitin chains create a ubiquitin code that is not completely deciphered yet [101]. K48- and K63-linked polyubiquitination represent the two most common mechanisms of ubiquitin linkage to protein substrates. The former is usually related to the proteolysis of the target protein via the ubiquitin-proteasome system (UPS) [101]. Conversely, the latter is usually involved in modulating the activity, interaction and intracellular trafficking of substrate proteins, participating in multiple biological processes, including cell growth and proliferation, apoptosis, DNA damage response, inflammation, neurodegenerative diseases and cancer [102].

Both of these distinct polyubiquitination mechanisms have been reported for Akt. K48-linked ubiquitination promotes the proteasomal degradation of Akt, thus turning off its activity and downstream responses. On the other hand, regarding the K63-linked polyubiquitination of Akt, multiple studies define its importance in the activation of Akt. Indeed, K63-linked ubiquitination serves as a regulatory signal that induces the plasma membrane recruitment, serine/threonine phosphorylation and activation of Akt (Figure 2).

### 2.1. K63-Linked Polyubiquitination of Akt

In 2009, Yang and collaborators showed the first evidence that Akt undergoes a K63-linked ubiquitination at the K8 and K14 residues within its PH domain in response to IGF-1 and cytokine Il-1 stimulation. TNF Receptor-Associated Factor 6 (TRAF6) has represented the first example of an E3 ligase that orchestrates Akt activation by inducing its K63-linked ubiquitination [54].

Since then, the list of E3-ubiquitin ligases promoting the K63-linked ubiquitination of Akt has been extended, suggesting that different extracellular signals utilize distinct E3-ubiquitin ligases to orchestrate Akt signaling pathway activation. These studies have also further demonstrated the involvement of K63-linked ubiquitination in several steps of cancerogenesis, including tumor onset, progression and invasion. In this regard, two distinct groups showed that S-Phase Kinase Associated Protein 2 (Skp2) and TNF Receptor Associated Factor 4 (TRAF4) promote Akt activation by mediating its K63-linked ubiquitination upon EGF stimulation. Notably, the ablation of E3-ubiquitin ligases of Akt TRAF6, Skp2 and TRAF4 impairs tumor progression in vivo [54,55,57]. Along this line, Skp2 depletion downregulates Akt activity, thus repressing Neu overexpression-driven breast cancer development [58]. Moreover, the subunit of SCF complex F-Box and Leucine Rich Repeat Protein 18 (FBXL18) promotes the glioma progression by inducing the K63-linked ubiquitination of Akt [103]. In agreement with the involvement of the K63-linked ubiquitination of Akt in cancerogenesis, the E3 ubiquitin ligase Ring Finger Protein 8 (RNF8) induces Akt activation by K63-linked ubiquitination, thus leading to lung cancer cell proliferation and chemotherapy resistance [103]. Furthermore, the SET domain bifurcate 1 (SETDB1)-mediated Akt K64 methylation induces K63-linked ubiquitination by several E3-ubiquitin ligases, including TRAF6 and Skp2-SCF complex, contributing to cancerogenesis [104]. The interaction of Lysine Demethylase 4B (KDM4B) with Akt stimulates the TRAF6-mediated K63-linked ubiquitination and activation of Akt, facilitating glucose metabolism and colorectal cancer growth [105]. Recent findings indicate that Cockayne Syndrome group A (CSA) is a novel Akt interacting protein that promotes its K63-linked ubiquitination and subsequent plasma membrane recruitment and activation. CSA depletion attenuates Akt phosphorylation and activation, thus leading to the diminished tumorigenic capabilities of breast cancer cells [106].

K63-linked ubiquitination seems to also have a critical role in regulating breast cancer invasion. Indeed, Ubiquitin-conjugating enzyme complex Ubc13-Uev1A promotes Akt signaling activation via K63-linked ubiquitination, thus leading to increased Cancer/Testis Antigen Family 45 Member A (CT45A) expression, cell migration and EMT signaling in breast cancer cells [107].

The physiological relevance of the K63-linked ubiquitination of Akt in cancerogenesis is further highlighted by the observation that the constitutively active and cancer-associated Akt E17K mutant is more effectively ubiquitinated by NEDD4-1 and shows a higher basal level of K63-linked ubiquitination than wild-type, thus explaining the increased activity and oncogenic potential of this mutant [54,108,109].

### 2.2. K48-Linked Polyubiquitination of Akt

Conversely, the involvement of the K48-linked ubiquitination of Akt in cancer can be attributed to its role in promoting its proteasomal degradation, thus resulting in shutting off the Akt pathway signaling. In this context, BReast CAncer gene 1 (BRCA1) plays a crucial role in negatively regulating Akt. Indeed, BRCA1 binds phosphorylated Akt and triggers its K48-linked ubiquitination and degradation. In agreement with this, BRCA1 deficiency, a condition commonly reported in several tumors, leads to the oncogenic activation of Akt. This emphasizes the involvement of the BRCA1-Akt interplay in tumorigenesis, suggesting the BRCA1-Akt pathway as a promising target in chemotherapies directed against BRCA1-deficient cancers [110]. Other E3 ligases have been further described to modify Akt by K48-linked ubiquitination, thus promoting the degradation and suppression of pro-tumorigenic Akt signaling [111,112,113,114,115]. TRIM13, for example, is an E3 ubiquitin ligase that has been found to be de-regulated in several tumor types, including B-cell chronic lymphocytic leukemia, multiple myeloma and non-small-cell lung carcinoma. TRIM13 may act as a tumor suppressor function by promoting Akt degradation, thus inducing p53 stabilization and apoptosis [112]. Moreover, MUL1 binds to Akt in its phosphorylated and active form to induce its degradation. Hence, MUL1 negatively modulates Akt signaling, regulating multiple cellular processes, including cell proliferation and migration [111]. Along this line, CHIP and TTC3 preferentially bind phosphorylated Akt and target it for proteasomal degradation, thus turning off Akt signaling and the Akt-related oncogenic signal transduction downstream [113,114]. In addition, the overexpression of ZNRF1, another E3 ubiquitin-ligase that has been described to induce the ubiquitination and degradation of Akt, has been associated with diminished protein levels of Akt and the subsequent inhibition of the proliferation and stemness properties of leukemia NB4 cells [115,116].

### 2.3. The Role of the Deubiquitination of Akt

Several studies also described the crucial role of different deubiquitinating enzymes (DUBs) in regulating the Akt signaling pathway. Indeed, DUBs such as CYLD, OTUD5 and USP1 can reverse the K63-linked ubiquitination of Akt and turn off its signaling activation [117,118,119]. Accordingly, the enhanced Akt ubiquitination and activation caused by the downregulation of OTUD5 gives rise to radioresistance in cervical cancer [119]. In line with this, the loss of CYLD promotes Akt hyperubiquitination and activation, as well as cell proliferation, survival and prostate tumorigenesis [120]. Noteworthily, the CYLD-mediated deubiquitination of Akt induced by bisdemethoxycurcumin has been shown to inhibit hepatocellular carcinoma cell growth [121]. Finally, a recent work suggests stimulating the de-ubiquitinating action of USP1, which is frequently de-regulated in multiple tumors, toward Akt as a putative therapeutic treatment of cancer [117].

## 3. Akt as a Target for Cancer Therapy

Due to its crucial involvement in carcinogenesis and drug resistance, Akt represents an attractive potential drug target for the development of anticancer therapies.

### 3.1. Targeting Akt Kinase

Nonselective Akt kinase inhibitors have been initially exploited, and medicinal chemistry efforts have been made to improve their pharmaceutical properties [122]. Nevertheless, the side effects observed in animal models limited their potential therapeutic application. More recently, many efforts have been made to achieve ATP-binding pockets structure-based Akt inhibitors with improved efficacy, selectivity and safety. Two ATP-competitive compounds, *capivasertib* and *ipatasertib*, have been extensively tested in clinical trials, and, to date, phase III clinical trials are under way for prostate and breast cancer [123,124].

In addition to directly targeting the kinase functionality of Akt, alternative therapeutic approaches have been pursued to control its aberrant activation. Among them, the phospholipid-containing molecule *perifosine,* by inhibiting the association of the PH domain with PIP3, has been shown to block Akt plasma membrane localization and its subsequent activation and phosphorylation [125]. Extensive phase I and II clinical programs on a large panel of cancer types are currently under way [126].

Additionally, the action of allosteric Akt inhibitors acting on the PH domain of Akt is exploited to maintain it in the inactive conformation. Currently, the structure-based design of small molecule agents that interact with various residues in the PH domains of Akt isoforms is exploited to allow for the isoform-specific inhibition of Akt [127]. In this context, an allosteric inhibitor named SC66 has been described, which interferes with the PIP3 binding function of the PH domain and, additionally, determines a robust K48-linked ubiquitination of Akt, which is finally targeted at the pericentrosomal region for proteasomal degradation. Along this line, the SC66 compound has shown anticancer activity in vitro and in vivo [128].

### 3.2. Targeting the Ubiquitin Pathway

A promising avenue for turning off the functionality of Akt is to interfere with the ubiquitination processes that target Akt. As we have seen in the previous paragraph, two distinct polyubiquitination modifications have been reported to regulate Akt signaling: K63-linked polyubiquitination, which is necessary for Akt membrane recruitment, phosphorylation and activation, and K48-linked polyubiquitination, which instead triggers the proteasomal degradation of phosphorylated Akt in order to silence its activity.

In light of this, a promising way to suppress the oncogenic activation of Akt might be to interfere with the K63 polyubiquitination. TRAF6 is the main E3 ligase that, in response to IGF-1 stimulation, promotes the K63 polyubiquitination of Akt. Interestingly, TRAF6 overexpression appears to be closely related to tumorigenesis and tumor development, and the analysis of TCGA and Gene Expression Omnibus (GEO) data indicates that the high expression of TRAF6 is significantly related to a poor prognosis compared with the low expression of TRAF6 [129]. In corroboration with these findings, the suppression of TRAF6 by shRNA impairs Akt activation in prostate cancer cells, suppresses tumor formation in the xenograft tumor model and potentiates apoptosis induced by chemotherapy agents [54]. Along this line, mir-124, miR-145 and miR-146 have been shown to repress TRAF6 protein translation and determine tumor suppressive effects on both primary and metastatic breast cancer [130,131,132,133,134,135]. Interestingly, it was shown that TRAF6, together with p62, K63-polyubiquitinates and activates mTOR [136]. Overall, TRAF6 can be considered an ideal therapeutic target for human cancer, and small molecules targeting TRAF6 may be considered as potential or adjuvant agents for cancer therapy. Targeting TRAF6 by inhibitors has been extensively studied. Proteasome inhibitors, such as bortezomib and MG132, have been shown to inhibit TRAF6 expression in myeloma and pancreatic cancer [137,138]. Small molecule inhibitors targeting TRAF6 have been developed: among them, C25-140 has been shown to inhibit, both in vitro and ex vivo, the production of K63-linked ubiquitin chains [139]. Epigallocate-chin-3-gallate (EGCG) is a novel E3 ubiquitin ligase inhibitor targeting TRAF6 assessed in melanoma [140]. Resveratrol has been shown to mediate the degradation of TRAF6 and decrease the proliferation and migration of prostate cancer cells [141]. Finally, the Chincona alkaloid has been shown to bind to the RING domain of TRAF6 and to promote the apoptosis of cancer cells [142].

Skp2 is the E3 ligase for ErbB/EGF2 receptor-mediated Akt K63-polyubiquitination, and its deficiency correlates with a decreased activation of Akt [56]. In corroboration, either the genetic or pharmacological targeting of Skp2 has been shown to hit cancer development in diverse genetic tumor models [143,144,145], thus pointing out that the Skp2 factor might represent a potential target for human cancer treatment. Consistently, one study shows that a small molecule (compound SZL-P1-41) that binds to Skp2 and prevents its binding with Skp1 disrupts Skp2 E3-ligase activity toward Akt, leading to the suppression of cancer progression in mouse models [146]. Interestingly, several substances, such as Longikaurin A, quercetin, Curcumin, lycopene, Rottlerin, nitidine chloride, Flavokawain and dioscin, have been shown to repress the expression of Skp2 in various types of human malignancies, thus arresting their proliferation [147].

### 3.3. Proteolysis-Targeting Chimeras

An alternative approach might be to increase K48-polyubiquitination to target Akt to the proteasome. The protein degraders Proteolysis Targeting Chimeras (PROTACs) are able to bind a E3 ligase protein and a target protein leading to its ubiquitination and proteasomal degradation [148]. These degraders could be exploited to degrade Akt as a potential therapy for cancer treatment. Along this line, the PROTAC degrader INY-03-041 has shown to degrade all the three Akt isoforms [149]. Other Akt de-graders, MS21 and MS143, which are von Hippel–Lindau (VHL)-recruiting PROTACs, shown to induce rapid and massive Akt degradation, thereby leading to the suppression of both cancer cells and tumor growth in vivo in a xenograft model [150].

## 4. Conclusions

The Akt pathway, one of the most frequently deregulated pathways in human cancer, affects a plethora of aspects of cancer malignancy. Mounting studies have provided deep knowledge of the mechanisms by which Akt is activated and inactivated as well as the diverse involvements of Akt in both tumor progression and drug resistance. This review was aimed to describe the multiple regulatory aspects of this broad signaling pathway with a particular focus on the multiubiquitination processes devoted to the Akt activation/deactivation cycles. A comprehensive understanding of this complex network will help to fully exploit the potential of small molecules acting on the inhibition or activation of this pathway. For example, an inhibitor of K63-linked Akt ubiquitination may be able to inhibit plasma membrane recruitment of Akt thus blocking its activation (Figure 3). On the other hand, activation of those E3 ligases that mediate K48-linked Akt ubiquitination might contribute to its inhibition. Therefore, targeting this pathway represents a unique and promising strategy that will push the field from science-based hypotheses to clinical applications and thereby hopefully reduce cancer-related mortality.

## Figures and Tables

**Figure 1 ijms-24-02809-f001:**
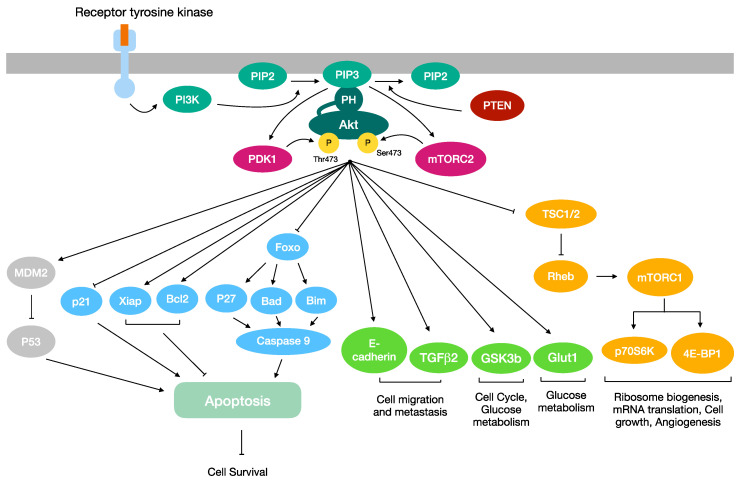
Overview of the PI3K/AKT/mTOR signaling pathway. Once phosphorylated, Akt translocates to a plethora of intracellular locations, where it modulates the function of numerous substrates. Akt may promote cell survival by regulating proteins involved in the activation of the apoptosis cascade. Akt activation stimulates the cell migration, cell cycle and glucose metabolism. Furthermore, it is a key regulator of angiogenesis, ribosome biogenesis, mRNA translation and cell growth.

**Figure 2 ijms-24-02809-f002:**
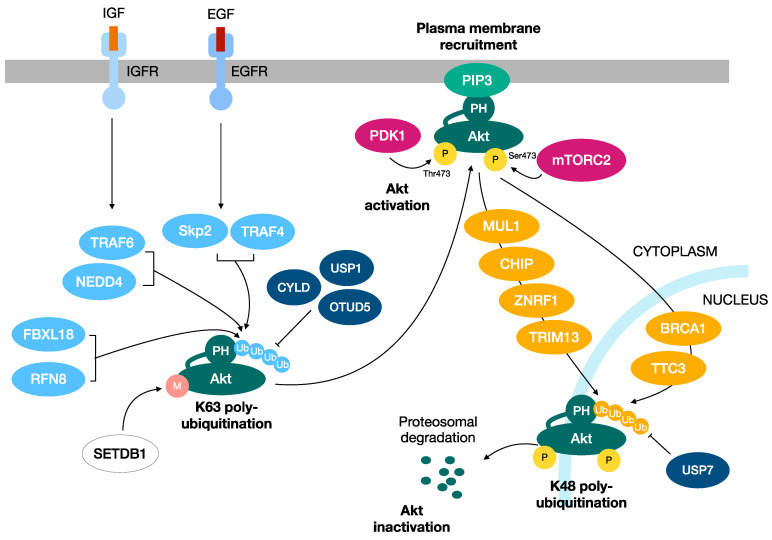
Two distinct ubiquitination processes regulate the activation and inactivation of Akt kinase. K48-linked ubiquitination promotes the proteasomal degradation of Akt, turning off its activity and downstream responses. Instead, the K63-linked polyubiquitination of Akt has a role in the activation of Akt, functioning as a regulatory signal that induces the plasma membrane recruitment, serine/threonine phosphorylation and activation of Akt.

**Figure 3 ijms-24-02809-f003:**
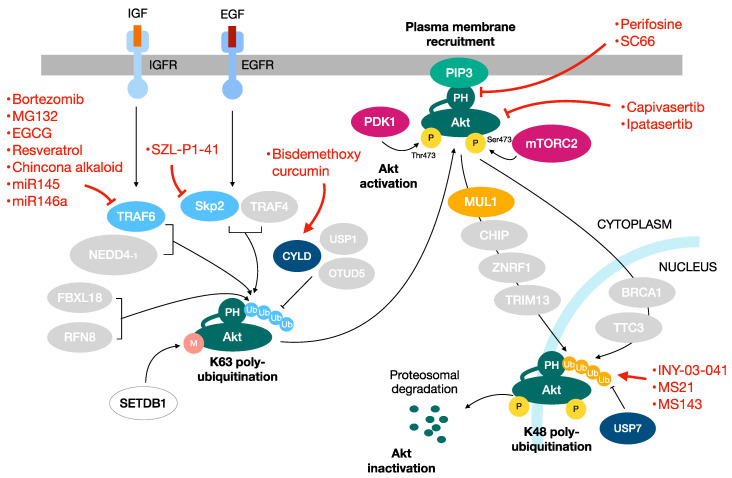
Targeting the Akt pathway in cancer. Akt is an attractive potential target for the development of anticancer therapies. Besides the drugs that directly target Akt functioning, other useful approaches are represented by the targeting of the ubiquitination pathways involved in Akt regulation. In red the drugs targeting the Akt pathway.

## Data Availability

No new data were created or analyzed in this study. Data sharing is not applicable to this article.

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
