# Peer review of "Interfering with the Ubiquitin-Mediated Regulation of Akt as a Strategy for Cancer Treatment"

_ijms, 2023, doi:10.3390/ijms24032809_

Round 1

Reviewer 1 Report

In this manuscript entitled “Interfering with ubiquitin-mediated regulation of Akt as a strategy for cancer treatment,” the authors summarize the ubiquitination-dependent regulation of Akt in cancer and its potential therapeutic strategies. Since increasing evidence has demonstrated the crucial roles of Akt ubiquitination in cancer, and the related signaling pathways are regarded as critical therapeutic targets, updating the knowledge in the research fields is beneficial to provide clinical insights. This is a comprehensive review and provides rich knowledge on the topic.

Comments:

Chapter 1

1.     Page 2: The authors describe Akt isoforms. The authors should elaborate on the distinct isoform-specific physiological roles, such as tissue distribution and knockout mouse phenotypes of each Akt isoform. 

2.     Page 2: The authors mention, “Akt kinase activity is finely tuned by post-translational modifications…”. The authors should update the knowledge of recently identified Akt PTMs, such as Akt hydroxylation and methylation, in cancer.

3.     Page 3: Ref [42] is incorrectly cited, which should be corrected to the following paper; PMID 15718470, “Phosphorylation and regulation of Akt/PKB by the rictor-mTOR complex.” Also, the authors need to clarify that mTORC2 is the responsible kinase for S473 phosphorylation. Ref [42] describes another layer of regulation of Akt activation through cell cycle-dependent S477/T479 phosphorylation by cyclin A/Cdk2. The description needs to be modified accordingly.

4.     Page 3: “Moreover, there are also PDK1 and mTORC2-independent mechanisms for Akt activation…” Ref [49-50], [51]. This sentence and citations are misleading. These papers are not for mechanistic studies on Akt activation; therefore, they are better to be removed.

5.     Page 3: “… to many downstream effectors, such as Glycogen Synthase Kinase 3 (GSK3), Forkhead transcription factor (FOXO) and Murine Double Minute 2 (MDM2) [41, 57, 58]”. MDM2 phosphorylation papers (PMID: 11504915, PMID: 11715018) are missing.

6.     The authors should cite the most relevant original-finding paper(s) instead of less relevant follow-up papers.

7.     Several grammatical errors need to be corrected.

Chapter 2

1.     Page 7: K48-linked ubiquitination is also critical in the Akt regulation. Sub-chapter 2.2, “K48-linked polyubiquitination of Akt,” should be expanded by discussing the details of the biological and pathological significance of Akt degradation mediated by each E3, CHIP, TTC3, TRIM13, ZNRF1, and MUL1, as was done in BRCA1-mediated degradation.

2.     Page 7: Akt regulation by deubiquitination is equally crucial as ubiquitination. This topic should be discussed more by elaborating on the biological and pathological roles of the reported deubiquitinases, CYLD, OTUD5, and USP1, in regulating Akt activity in cancer. It would be beneficial to expand the paragraph and discuss it in a separate sub-chapter, “2.3”, rather than being discussed in chapter 3.

Chapter 3

1.     The chapter should be split into three sub-chapters: e.g. (3.1.) Targeting Akt kinase; (3.2.) Targeting ubiquitin pathway; (3.3.) PROTACs.

2.     It would be helpful to insert a Table that displays inhibitors, signaling modifiers, degraders, or therapeutic intervention strategies.

Author Response

Please, consider our answer point by point:

Chapter 1

  1. Page 2: The authors describe Akt isoforms. The authors should elaborate on the distinct isoform-specific physiological roles, such as tissue distribution and knockout mouse phenotypes of each Akt isoform.

We took into account reviewer's concern (see text in red).

  1. Page 2: The authors mention, “Akt kinase activity is finely tuned by post-translational modifications…”. The authors should update the knowledge of recently identified Akt PTMs, such as Akt hydroxylation and methylation, in cancer.

We added the references.

  1. Page 3: Ref [42] is incorrectly cited, which should be corrected to the following paper; PMID 15718470, “Phosphorylation and regulation of Akt/PKB by the rictor-mTOR complex.” Also, the authors need to clarify that mTORC2 is the responsible kinase for S473 phosphorylation. Ref [42] describes another layer of regulation of Akt activation through cell cycle-dependent S477/T479 phosphorylation by cyclin A/Cdk2. The description needs to be modified accordingly.

We corrected this reference.

  1. Page 3: “Moreover, there are also PDK1 and mTORC2-independent mechanisms for Akt activation…” Ref [49-50], [51]. This sentence and citations are misleading. These papers are not for mechanistic studies on Akt activation; therefore, they are better to be removed.

We removed the sentence.

  1. Page 3: “… to many downstream effectors, such as Glycogen Synthase Kinase 3 (GSK3), Forkhead transcription factor (FOXO) and Murine Double Minute 2 (MDM2) [41, 57, 58]”. MDM2 phosphorylation papers (PMID: 11504915, PMID: 11715018) are missing.

We added the references.

  1. The authors should cite the most relevant original-finding paper(s) instead of less relevant follow-up papers.

We cited most relevant papers.

  1. Several grammatical errors need to be corrected.

We fixed this point

Chapter 2

  1. Page 7: K48-linked ubiquitination is also critical in the Akt regulation. Sub-chapter 2.2, “K48-linked polyubiquitination of Akt,” should be expanded by discussing the details of the biological and pathological significance of Akt degradation mediated by each E3, CHIP, TTC3, TRIM13, ZNRF1, and MUL1, as was done in BRCA1-mediated degradation.

We took into account reviewer's concern (see text in red).

  1. Page 7: Akt regulation by deubiquitination is equally crucial as ubiquitination. This topic should be discussed more by elaborating on the biological and pathological roles of the reported deubiquitinases, CYLD, OTUD5, and USP1, in regulating Akt activity in cancer. It would be beneficial to expand the paragraph and discuss it in a separate sub-chapter, “2.3”, rather than being discussed in chapter 3.

We took into account reviewer's concern (see text in red).

Chapter 3

  1. The chapter should be split into three sub-chapters: e.g. (3.1.) Targeting Akt kinase; (3.2.) Targeting ubiquitin pathway; (3.3.) PROTACs.

We took into account reviewer's concern (see text in red).

  1. It would be helpful to insert a Table that displays inhibitors, signaling modifiers, degraders, or therapeutic intervention strategies.

We decided to not insert a table because we think that is out of the scope of this review mostly focussed on ubiquitination pathway as a target. 

Reviewer 2 Report

The authors have assembled an informative review on the role of Akt kinases in the regulation and de-regulation of cells, and how targeting ubiquitylation of Akt might be used in cancer therapy. Even though I am unable to verify the accuracy of all citations, I therefore believe that this review is worthy of publication in IJMS, after some smaller corrections and adjustments (see below).

Figures should have a legend describing what is shown. In the text, there is only one mentioning of Fig. 1 in lines 24-28: “Indeed, phosphorylated Akt (pAkt) translocates to a plethora of intracellular locations, where it modulates the function of numerous substrates, many of them being involved in cell proliferation and growth, metabolism, angiogenesis, resistance to hypoxia and migration [7-9] (Fig. 1)”. Apart from “metabolism”, there is not much shown in this figure that relates to this statement. Instead, one could several times refer to this figure in later paragraphs. In general, there are too few hints towards to all the figures in the main text.

Lines 11-12: "Akt is frequently deregulated in many types of human cancers, its overexpression or abnormal activation being associated with increased cancer cell proliferation and survival

In this sentence in the abstract, one could get the false idea that the authors talk about “survival” of a cancer patient. Maybe, by changing the order of words on could avoid such a misunderstanding: e.g.: “…increased proliferation and survival of cancer cells

Line 15: There is no plural of “evidence”.

Line 22: “The PI3K/Akt/mTOR pathway (insert: was or has been) shown to be…”

Line 31: “contribution

Line 41: I do not think a discovery reported in 2009 classifies as “recent”.

Lines 66-67: …” sharing (delete “the”) 90-95% of homology in the kinase region and the (delete “the”) 60% of homology

Line 143:  “In addition..” instead of “In addiction…”

Lines 161-162: “…role in EMT, a hallmark of which is the downregulation of E-cadherin”

Line 170: “…has been shown to modulate (delete s) the metastatic…”

Lines 177-178: “…instead de-regulated Akt function supports abnormal proliferation and survival of cancer cells."

Lines 186-187: “Akt may alter the expression of Glucose Transporter 1 (GLUT1) by increasing translation of its mRNA through mTORC1”

Lines 189-190: “…metabolism, since it is the kinase responsible…”

Line 197: “…in a process defined as ubiquitination”

Line 200: “The covalent addition…”

Line 211: “…linked to each other…”

Lines 212-213: “K48- and K63-linked polyubiquitination represent the two most common mechanisms… “ 

Lines 268-272: According to the instruction for authors of IJMS “In Reviews:  Unpublished research data are not allowed.

Lines 296-300: The description of the roles of K63-specific Dubs on Akt regulation seems misplaced under the headline of the paragraph: “2.2. K48-linked polyubiquitination of Akt

Line 348: „Proteasome inhibitors…“

Line 353: „Resveratrol was (or has beenshown…” The same problem occurs in line 359, and line 366 (“were or have been shown”, and line 373 (“was shown” or “has been shown”)...

Author Response

We took into account all the reviewer's concerns (see text in red). In particular we implemented the figures' legends.